# Detection of M2-Type Anti-Mitochondrial Autoantibodies against Specific Subunits in the Diagnosis of Primary Biliary Cholangitis in Patients with Discordant Results

**DOI:** 10.3390/diagnostics13111840

**Published:** 2023-05-24

**Authors:** Cristina Arnaldos-Pérez, Albert Pérez-Isidro, Uma Bolos, Carmen Domènech, Judit Ballús, Sergio Rodríguez-Tajes, María Carlota Londoño, Odette Viñas, Estíbaliz Ruiz-Ortiz

**Affiliations:** 1Department of Immunology, Centre de Diagnòstic Biomèdic, Hospital Clínic de Barcelona, Villarroel 170-Escala 4, Planta 0, 08036 Barcelona, Spain; arnaldos@clinic.cat (C.A.-P.); alperezi@clinic.cat (A.P.-I.); bolos@clinic.cat (U.B.); mcdomen@clinic.cat (C.D.); jballus@clinic.cat (J.B.); ovinyas@clinic.cat (O.V.); 2Institut de Recerca Biomèdica August Pi i Sunyer (IDIBAPS), 08036 Barcelona, Spain; srodriguez@clinic.cat (S.R.-T.); mlondono@clinic.cat (M.C.L.); 3Liver Unit, Hospital Clínic de Barcelona, 08036 Barcelona, Spain; 4Centro de Investigación Biomédica en Red de Enfermedades Hepáticas y Digestivas (CIBERehd), Instituto de Salud Carlos III, 28029 Madrid, Spain

**Keywords:** primary biliary cholangitis, AMA-M2, E2, PDC, BCOADC, OGDC, indirect immunofluorescence, ELISA, Dot-blot, autoantibodies

## Abstract

Background: M2-type anti-mitochondrial autoantibodies are considered the hallmark of primary biliary cholangitis and are directed mainly against the E2 subunits of the 2-oxo acid dehydrogenase complex enzymes (PDC, BCOADC and OGDC). The aim of this study was to determine whether a Dot-blot that includes these E2 subunits separately could confirm the results of methods with non-separated subunits in patients with low positive or discordant results between techniques. Methods: Sera of 24 patients with low positive or discordant results and of 10 patients with clear positive results by non-separated subunits methods were analyzed by Dot-blot with separated subunits. Results: Autoantibodies against E2 subunits of PDC, BCOADC or OGDC were detected in all patients, except in one case from the low positive or discordant results group, by Dot-blot with separated subunits. Conclusions: It would be advisable to use methods that include the three E2 subunits, and a Dot-blot with separated subunits could confirm doubtful cases by non-separated assays.

## 1. Introduction

Primary biliary cholangitis (PBC) is a chronic autoimmune liver disorder characterized by inflammation of intrahepatic biliary ducts, which leads to fibrosis and consequently may produce cirrhosis and liver failure [1,2,3,4,5]. Its prevalence is low and it mostly affects middle-aged women [1,6]. The etiology is unclear, but many data suggest that a genetic predisposition along with environmental risk factors such as infections or smoking may be a crucial responsible [1,4]. Clinical presentation is variable, and patients range from asymptomatic and stable or slowly progressive to symptomatic and rapidly progressive [3]. The prognosis depends on the development of liver cirrhosis and its complications [2].

Diagnosis of PBC can be established if two out of the three following criteria are met: sustained elevated levels of alkaline phosphatase (ALP), evidence of antimitochondrial autoantibodies (AMA) or specific antinuclear autoantibodies (ANA) and diagnostic liver histology [4]. Therefore, a biopsy can be avoided in case of high ALP levels and detection of autoantibodies [2].

A total of nine types of anti-mitochondrial autoantibody patterns (M1 to M9) have been described, which are associated with different antigens and with hepatic diseases but also with non-hepatic disorders [1,2]. Only M2-type AMA (AMA-M2), which are strongly associated with PBC, are considered the hallmark of the disease. They are detected in 90–95% of patients, whereas specific ANAs, namely, Sp100 and gp-210, are only present in 50–70% of patients with PBC and in 30–50% of AMA-M2-negative PBC patients [1,2,4,5,7]. AMA-M2 target components of the 2-oxo-acid dehydrogenase complex: pyruvate dehydrogenase complex (PDC), 2-oxoglutarate dehydrogenase complex (OGDC) and branched-chain 2-oxoacid dehydrogenase complex (BCOADC). Specifically, autoantibodies mainly recognize the E2 subunits of these complexes: PDC-E2 (80–90% of cases), BCOADC-E2 (50–80% of cases) and OGDC-E2 (20–60% of cases), and to a lesser extent, the E1 and E3 subunits [2,5,6].

The accurate detection of AMA-M2 is relevant as it is considered the hallmark of the disease. Its recognition could avoid a biopsy and lead to an earlier diagnosis, which is essential to start an adequate treatment [5,6]. Traditionally, IFA has been the most accepted assay for its detection; however, several antigen-specific methods such as ELISA or Dot-blot (DB) have been developed. These assays use a fusion protein combining the three E2 subunits, a mixture of recombinant E2 subunits or the three E2 recombinant subunits isolated, among others [5,6,8].

ELISA and DB appear to be more sensitive than IFA [4]. Poyatos et al. [2] reported a group of patients with a negative IFA but with the presence of several combinations of autoantibodies against the three different E2 subunits by DB. However, further studies were not performed to determine whether they were false-positive or false-negative results.

Our group previously published a study in which sera of 17 patients with liver diseases were analyzed by 40 laboratories, each with its diagnostic algorithm. For some patients, a low percentage of laboratories reported AMA-M2 pattern by IFA; however, this percentage was clearly superior when samples were analyzed by antigen-specific techniques using the three E2 subunits together or separately [6].

To date, no studies have compared the application of antigen specific assays with the three E2 subunits together and those that have them separated. Furthermore, the use of one of these type of assays to clarify discordant results between other techniques is unclear.

In our laboratory, we analyze serum samples by indirect immunofluorescence assay (IFA) on rat triple tissue and by antigen-specific techniques that use the three E2 subunits together; that is to say, antigen is a combination of subunits. The aim of this study was to evaluate whether, in doubtful cases due to low positive or discordant results between assays used in our laboratory, the use of a DB with separated subunits (DB-E2sep) could confirm the presence of AMA-M2, ruling out the possibility that it is a false-positive result.

## 2. Patients and Methods

### 2.1. Patients

We included 34 patients who had been evaluated by all our routine assays: 24 patients with low positive or discordant results between these techniques (group A) and 10 patients with a clear positive result in all of them (group B), as a positive control group (Appendix A).

Clinical and laboratory data were collected (Appendix A). We analyzed sera samples available in our facilities, which could belong to the first diagnostic study or to follow-up; therefore, some patients were under treatment. This study was conducted in accordance with the Declaration of Helsinki and was approved by the Research Ethics Committee of the Hospital Clínic de Barcelona (HCB/2022/0923).

### 2.2. Laboratory Assays

#### 2.2.1. Routine Assays (Non-Separated Three E2 Subunits)

Routine assays used to assess AMA were IFA, enzyme-linked immunosorbent assay (ELISA) and DB. All of them were performed according to the manufacturer’s instructions. IFA (NOVA Lite^®^ Rat Liver, Kidney, Stomach Kit, Inova, San Diego, CA, USA. Ref. 704180) was applied on rat kidney, stomach and liver sections (Rat Triple Tissue; RTT) at 1:40 dilution. ELISA assay (QUANTA Lite^®^ (MIT3), Inova, San Diego, CA, USA. Ref. 704540) contained a purified recombinant antigen with the immunodominant portions of PDC-E2, BCOADC-E2 and OGDC-E2 (3E2). The cut-off recommended by the manufacturer is 20 U/mL. DB assay (Euroimmun, Lübeck, Germany. Ref. DL 1300-1601-4 G) (DB-E2nonSep) included a recombinant fusion protein comprising the immunogenic domains of the three E2 subunits (3E2) and an antigen natively purified from bovine heart with PDC-E2 as main component (nAg (mainly PDC-E2)). Results were analyzed by EUROLineScan software (Ref. YG 0006-0101 version 3.4.36) to quantify the band intensity. In group A, we included patients with a negative or low positive result for nAg (mainly PDC-E2) and a positive result for 3E2.

#### 2.2.2. Dot-Blot with the Three E2 Subunits Separated (DB-E2sep)

Every sample was analyzed by a DB that evaluated autoantibodies against each E2 subunit separately (D-Tek, Mons, Belgium. Ref. LI10D-24). This DB-E2sep contained recombinant (1) PDC-E2, (2) BCOADC-E2 and (3) OGDC-E2 and (4) native E1, E2 and E3 subunits of PDC purified from bovine heart (nPDC). A visual (qualitative) interpretation of the results was performed. The Positive Control dot must be positive for each strip. Then, a comparison between the intensity of the specific antigen dots and the Negative Control dot was performed, given the lack of an automatic software.

### 2.3. Statistical Methods

Statistical analyses were performed using GraphPad Prism 9.5.0. Differences between groups in quantitative variables were assessed using Mann–Whitney test after evaluating the normality of populations using D’Agostino–Pearson normality test. Differences in proportions between groups were assessed using Fisher’s exact test.

## 3. Results

### 3.1. Biochemical and Immunological Results and Diagnostics Groups

Overall, in group A and B, 87% and 70% were women, and mean age was 62.8 and 64.7%, respectively. Data of biochemical and immunological variables of the studied samples were collected (Table 1). In both groups, the median of all variables was within normal range. IgM levels were significantly higher in group B as well as number of patients with high IgM results (2/22; 9.1% in group A vs. 5/10; 50.0% in group B). However, no differences were observed in the other studied parameters.

Groups did not show any differences in the proportion of patients in each diagnostic category. However, there was a non-significant trend toward a higher proportion of patients with PBC in group B (33.3% vs. 70.0%, *p* = 0.26) (Table 2).

### 3.2. Detection of Autoantibodies by Several Assays

IFA results of patients in our cohort are shown in Figure 1. Clearly positive AMA-M2 results (>1:160) by IFA were obtained in all patients (100.0%) from group B, whereas negative (62.5%), doubtful positive (16.7%) or low positive (20.8%) results were obtained in group A. Only 12.5% of patients in group A were clearly positive. ELISA assay was positive in all patients in both groups, but mean concentration was 68.2 and 129.7 U/mL in group A and B, respectively (*p* < 0.01). Autoantibodies against 3E2 in DB-E2nonSep were quantified displaying a tendency of higher results in group B (40.0 vs. 208.5; *p* < 0.01).

Each group was also studied with a DB-E2sep. We detected autoantibodies anti-nPDC in 25.0% (6/24) vs. 100.0% (10/10), anti-PDC-E2 in 25.0% (6/24) vs. 100.0% (10/10), anti-BCOADC-E2 in 70.8% (17/24) vs. 80.0% (8/10) and anti-OGDC-E2 in 8.3% (2/24) vs. 30.0% (3/10) of cases in group A and B, respectively (Table 3 and Figure 2). Although percentage of patients with autoantibodies against nPDC (6/24; 25.0%) and PDC-E2 (6/24; 25.0%) by DB-E2sep was the same in group A, only two patients had autoantibodies against both antigens (Figure 2).

Percentage of patients with different combinations of autoantibodies against specific subunits is shown in Table 3. Despite the prevalence of autoantibodies anti-BCOADC-E2, being similar between groups (70.8% in A and 80.0% in B, *p* = 0.69), the presence of these autoantibodies alone was observed only in group A (54.2% vs. 0%, *p* < 0.01).

### 3.3. Comparison between Routine Techniques (Non-Separated Subunits) and Dot-Blot with Separated Subunits

Results of all assays performed on the samples are shown in Figure 2.

All patients in both groups, except one in group A, had autoantibodies against at least one subunit by DB-E2sep. The only case without autoantibodies in this assay had the following results in routine techniques: negative by IFA, weak positive by ELISA (3E2) (22.8 U/mL) and negative autoantibodies anti-nAg (mainly PDC-E2) and moderate-high positive autoantibodies anti-3E2 by DB (DB-E2nonSep).

Patients in group A were included because of a negative or low positive result for autoantibodies anti-nAg (mainly PDC-E2) of DB-E2nonSep. However, in some cases, autoantibodies anti-nPDC or anti-PDC-E2 were detected in DB-E2sep. The only case with low positive autoantibodies anti-nAg (mainly PDC-E2) (first row, Figure 2) also displayed a positive result in nPDC of DB-E2sep.

We did not observe any differences to point out between the results of IFA (positive, probably positive, or negative) and the presence or absence of autoantibodies against a specific subunit.

## 4. Discussion

PBC is a progressive disorder that can lead to cirrhosis [1,3,4,5]. Its diagnosis is relevant as treatment can delay progression in most cases [3]. The presence of AMA-M2 is included in the diagnostic criteria. These autoantibodies are detected in 90–95% of patients with PBC and could avoid the need for a biopsy [4]. Therefore, an accurate detection is important.

Currently, there is a wide variety of available techniques and commercial kits that allow us to identify the presence of AMA-M2 autoantibodies. However, there is no consensus on the most appropriate methods that should be used to detect these autoantibodies. In our laboratory, in addition to AMA-M2 detection by IFA on RTT, we perform antigen-specific assays that use the nPDC-E2 (DB) and recombinant 3E2 antigen (ELISA and DB) to study the presence of AMA-M2. We hypothesized if, in doubtful cases due to a low positive or discordant results between techniques, the use of a DB-E2sep would confirm the presence of these autoantibodies detected, ruling out the possibility that it is a false-positive result.

Based on this aim, we selected a group of patients with discordant results between IFA and antigen-specific techniques for the determination of AMA-M2 autoantibodies (group A), all of whom were positive for 3E2 by DB-E2nonSep (most of them with low levels). We then compared this group with a control group (group B), which included patients with clearly positive results for all the different techniques included in the study. In relation to biochemical and immunological variables in our population, median levels of all parameters were within normal range in both groups, probably because of the various diagnoses and treatment in some patients (A/4, A/5, A/7 and A/9 were under treatment with immunosuppressors, and A/13, A/18, B/1, B/2, B/5, B/6, B/7 and B/8 were treated with ursodeoxycholic acid). However, the median IgM level was higher in group B. In addition, there were no differences in diagnosis between groups although a trend toward a higher proportion of patients with PBC in group B was observed. Nevertheless, the detection of AMA-M2 in group A could make necessary a closer follow-up of these patients.

As for biopsy, in patients with positive AMA-M2 and normal ALP levels, there is no indication for liver biopsy. In our center, we follow up these patients (A/14, A/15, A/21, A/23 (Appendix A)) as they could develop PBC over time [7]. The detection and pattern of autoantibodies should be assessed alongside with clinical findings. The prevalence of AMA-M2-positive cases in the general population is considerable (up to 1/1000), whereas PBC prevalence is modest (up to 0.4/1000). Several studies imply that only one out of every six patients with AMA-M2-positive and normal ALP levels develops PBC within five years [9]. On the other hand, other studies found that in patients with positive AMA-M2 autoantibodies and normal ALP levels, the biopsy revealed histological PBC characteristics in 80% of them [10]. A biopsy will provide more information, although it is invasive and should be performed on an individual basis.

Regarding routine AMA-M2 evaluation, we compared the results of assays using 3E2 (DB-E2nonSep and ELISA), nPDC-E2 (DB-E2nonSep) and IFA. Thereafter, in order to confirm the discordant results in group A, we studied the sera of all patients (groups A and B) with DB-E2sep. Significant differences between both groups were observed in relation to IFA by RTT (Figure 1). In group B, all patients (100%) showed clearly positive results (AMA-M2 pattern with titers ≥ 320), whereas in group A, only 12.5% of patients presented a pattern consistent with the presence of AMA-M2 but at a low titer (≤160). Moreover, 62.5% had negative IFA results. Despite these data, all patients (100% in group A and 100% in group B) were positive via ELISA (3E2). Nevertheless, differences were observed in relation to median values between groups, being higher in patients from group B (68.2 vs. 129.7 U/mL; *p* < 0. 01). ELISA and DB assays using recombinant proteins with 3E2 subunits have been reported to be more sensitive than IFA [11]. Other studies have detected these autoantibodies using ELISA with 3E2 subunits in patients with negative IFA results [5]. Therefore, it has also been suggested that IFA should not be the only first-line assay, but should be used in combination with other assays, such as ELISA or DB [4,6]. In fact, if our cohort had only been studied by IFA, the 15 patients with negative IFA in group A (62.5%) would have obtained an incorrect result. This would have made a biopsy necessary in the case of patients with elevated ALP levels and a lack of specific antinuclear autoantibodies. However, as in our center we always perform both IFA and antigen-specific assays, patients included in group A were followed up closer because they could develop PBC at some point [7]. Regarding the results obtained with the DB-E2nonSep (routinely used in our laboratory) all patients were positive for 3E2 antigen (because of inclusion criteria) but with differences between groups in relation to the level of positivity (higher results in group B). Finally, all samples were studied by a DB-E2sep in order to clarify discordant results. In our cohort the results for autoantibodies anti-nPDC, PDC-E2, BCOADC-E2 and OGDC-E2 were 25.0%, 25.0%, 70.8% and 8.3% in group A and 100.0%, 100.0%, 80.0% and 30.0% in group B, respectively. Poyatos et al. [2] performed the same kit (DB-E2sep) in their cohort of AMA-M2 positive patients, regardless of diagnosis, and obtained a prevalence of 83.0%, 65.0%, 44.0% and 10.0%, respectively. The frequencies in our group B (clearly positive patients) were more similar to those reported by Poyatos et al. [2] than those in our group A. This may be due to the fact that we included patients with the most clear and higher values of AMA-M2 in group B. However, they do not represent all usual patients. In our study, all patients (100%) in group B had anti-nPDC and PDC-E2 autoantibodies, which are more frequently detected by most of the available methods [2]. However, anti-BCOADC-E2 were the most prevalent autoantibodies in group A, in some cases together with PDC positivity but in other cases alone. Nevertheless, in some patients in group A, autoantibodies anti-nPDC or anti-PDC-E2 were detected by DB-E2sep but not by nAg (mainly PDC-E2) included in DB-E2non-Sep, showing that autoantibodies detection may present discrepancies between different commercial products even when similar antigens are used. Cases with positive nPDC and negative PDC-E2 by DB-E2sep and negative nAg (mainly PDC-E2) by the DB-E2nonSep, could be due to the fact that nPDC includes E1, E2 and E3 subunits but PDC-E2 and nAg (mainly PDC-E2) include only the E2 subunit or at least is more represented. On the other hand, four of these negative cases for autoantibodies anti-nAg (mainly PDC-E2) by DB-E2nonSep, had only autoantibodies against the recombinant antigen (PDC-E2 of DB-E2sep), possibly due to the recognition of other epitopes. The only low positive case for autoantibodies anti-nAg (mainly PDC-E2) by DB-E2nonSep was confirmed by the native component (nPDC) of DB-E2sep but not by PDC-E2.

The aim of this study was to ascertain whether DB-E2sep could confirm the results of doubtful cases by our routine techniques (non-separated subunits). In all patients in both groups, except for one in group A, autoantibodies against at least one subunit were detected by DB-E2sep.

We did not observe a clear relationship between IFA results and the presence of autoantibodies against a specific subunit, probably because of the limited size of our cohort. However, Poyatos et al. [2] showed a significant association between IFA titer and the frequency of autoantibodies against a specific subunit or a combination of them.

Another study, in which samples of 17 patients were analyzed by 40 laboratories according to their working algorithm, showed that patients with autoantibodies anti-PDC-E2 were detected by IFA in all participant laboratories. However, when autoantibodies anti-BCOADC-E2 were present alone, IFA performance between centers was variable [6]. In our cohort, patients with autoantibodies anti-BCOADC-E2 alone showed negative (7/24), doubtful positive (3/24) and consistent positive AMA-M2 pattern (3/24) results by IFA. However, some patients with autoantibodies anti-PDC-E2 also showed negative IFA results. These differences, in contrast with the previously reported, could be due to the small cohort and the bias of the inclusion criteria. Nevertheless, the confirmation of results in our study by DB-E2sep highlights the importance of using at least one method that includes the three E2 subunits apart from IFA and matches the published results.

In the present study, the antigen-specific assays that we performed and included the three E2 subunits were ELISA (Werfen) and DB (Euroimmun and D-Tek). To our concern, several companies provide similar products including the three E2 subunits: Euroimmun, Werfen, D-Tek, Alpha Dia, Orgentec and Human Imtech.

The main limitation of this study was, as mentioned, the small number of enrolled patients. Large population studies including patients with similar characteristics to those of group A and performing assays with the three E2 subunits together and separated are required to confirm these results. In addition, it would make it possible to analyze the prevalence of autoantibodies or their combination between diagnostic groups. However, this study allowed us to confirm that these patients with discordant results between techniques indeed had autoantibodies.

## 5. Conclusions

According to our study, DB-E2sep has been able to confirm all cases of low positive or discordant results except one. However, further studies with larger sample sizes should be performed.

Our final recommendation is that it would be advisable to use methods, apart from IFA, which include the three E2 subunits, either separated or combined, and not only PDC, as patients could only have autoantibodies against other E2 subunits (BCOADC and OGDC).

## Figures and Tables

**Figure 1 diagnostics-13-01840-f001:**
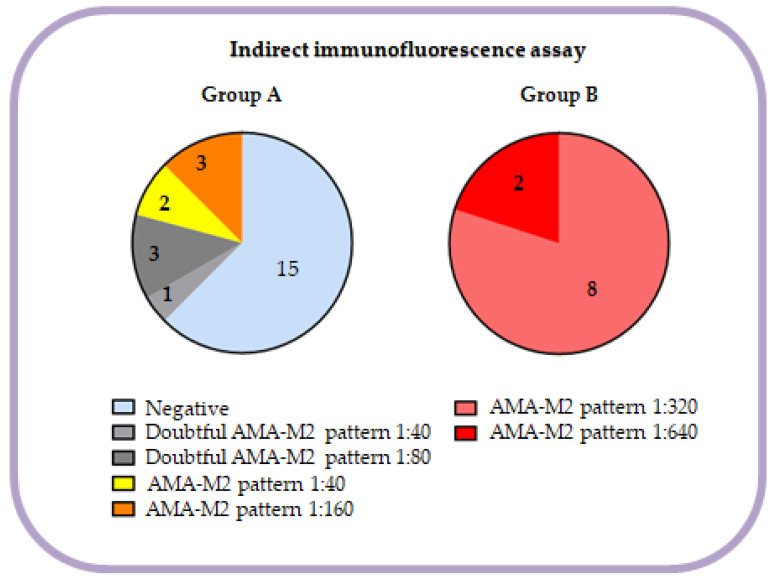
Distribution of IFA results of every patient in each group.

**Figure 2 diagnostics-13-01840-f002:**
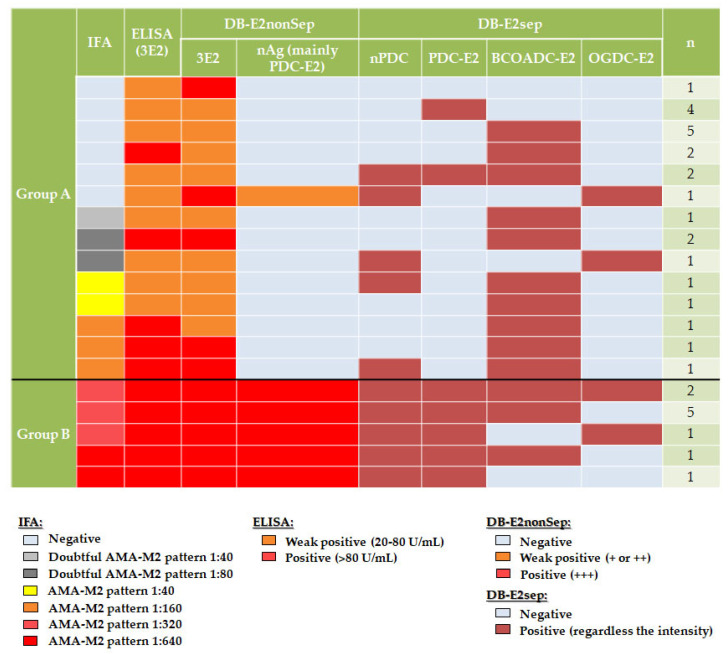
Results of AMA-M2 assays performed in both groups.

**Table 1 diagnostics-13-01840-t001:** Biochemical and immunological results of both groups.

	Group A (*n* = 24) (Median) (min–max)	Group B (*n* = 10) (Median) (min–max)	*p* Value
GGT [5–40]	44 U/L (14–262) (*n* = 23)	35 U/L (13–165)	0.33
ALP [46–116]	104 U/L (44–391)	97 U/L (68–238)	0.72
AST [5–40]	30 U/L (15–501)	28 U/L (21–44)	0.79
ALT [5–40]	31 U/L (<7–722)	32 U/L (15–58)	0.85
Total bilirubin [<1.2]	0.7 mg/dL (0.3–6.1)	0.6 mg/dL (0.4–0.9)	0.82
IgM [0.36–2.60]	1.45 g/L (0.65–4.60) (*n* = 22)	2.65 g/L (1.20–5.79)	0.01 *
IgG [6.8–15.3]	12.7 g/L (6.5–44.3) (*n* = 22)	11.8 g/L (8.6–28.7)	0.68
IgA [0.66–3.65]	2.71 g/L (0.97–7.41) (*n* = 22)	2.07 g/L (1.31–2.97)	0.18

GGT: Gamma-Glutamyl Transferase; ALP: Alkaline Phosphatase; AST: Aspartate Aminotransferase; ALT: Alanine Aminotransferase. * *p* < 0.05. Values between “[]” indicate normal laboratory values.

**Table 2 diagnostics-13-01840-t002:** Diagnosis of each patient at the moment of the study.

	Group A (*n* = 24)	Group B (*n* = 10)	*p* Value
All PBC	8 (33.3%)	7 (70.0%)	0.07
Only PBC	7 (29.2%)	4 (40.0%)	0.69
Overlap PBC/AIH	1 (4.2%)	3 (30.0%)	0.07
Probable PBC	1 (4.2%)	0 (0.0%)	>0.99
Other diagnoses	15 (62.5%)	3 (30.0%)	0.13
Not PBC criteria	11 (45.8%)	3 (30.0%)	0.47
Only AIH	4 (16.7%)	0 (0.0%)	0.3

AIH: Autoimmune hepatitis; PBC: Primary biliary cholangitis.

**Table 3 diagnostics-13-01840-t003:** Observed combinations of autoantibodies against specific subunits.

Autoantibodies against	Group A (*n* = 24)	Group B (*n* = 10)	*p* Value
PDC-E2 (only)	4 (16.7%)	0 (0.0%)	0.30
BCOADC-E2 (only)	13 (54.2%)	0 (0.0%)	<0.01 *
nPDC/PDC-E2	0 (0.0%)	1 (10.0%)	0.29
nPDC/BCOAD-E2	2 (8.3%)	0 (0.0%)	>0.99
nPDC/OGDC-E2	2 (8.3%)	0 (0.0%)	>0.99
nPDC/PDC-E2/BCOAD-E2	2 (8.3%)	6 (60.0%)	<0.01 *
nPDC/PDC-E2/OGDC-E2	0 (0.0%)	1 (10.0%)	0.29
nPDC/PDC-E2/BCOAD-E2/OGDC-E2	0 (0.0%)	2 (20.0%)	0.08
No autoantibodies	1 (4.2%)	0 (0.0%)	>0.99

* *p* < 0.05.

## Data Availability

Anonymized data not published within this article will be made available by request from qualified investigators.

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
