# Peer review of "Detection of M2-Type Anti-Mitochondrial Autoantibodies against Specific Subunits in the Diagnosis of Primary Biliary Cholangitis in Patients with Discordant Results"

_diagnostics, 2023, doi:10.3390/diagnostics13111840_

Round 1
Reviewer 1 Report
The manuscript entitled " Detection of M2-type Anti-mitochondrial Autoantibodies against Specific Subunits in the Diagnosis of Primary Biliary Cholangitis in Patients with Discordant Results" presented by Arnaldos-Pérez et al, summaries a research pertaining to M2-type Anti-mitochondrial Autoantibodies as a diagnostic tools for Primary Biliary Cholangitis. However, some minor exercise is required to improve further its content:
· Authors need to more emphasize on current literature and work done on this area in Introduction.
· Any other future studies that need to warranted must be concluded for conversion of these findings.
· Authors need to conclude the results with reference to available market kits
· Language and any other typological mistake can be address
· Please check pattern of reference as per format.
Author Response
First of all, thank you for your time in reviewing our paper. We have carefully considered your comments and tried our best to improve our current version according to them.
1) Authors need to more emphasize on current literature and work done on this area in Introduction
According to your suggestion, we have modified the introduction with work done in this area recently explaining why we decided to carry out this study. (Lines 62 -75)
2) Any other future studies that need to warranted must be concluded for conversion of these findings.
In this regard we have modified the last part of Discussion. (Lines 287 - 290)
3) Authors need to conclude the results with reference to available market kits
According to your suggestion, we have mentioned in the last part of Discussion companies that, to our concern, provide similar kits. (Lines 282 - 285)
4) Language and any other typological mistake can be address
We have made several changes throughout the paper.
5) Please check pattern of reference as per format.
We have revised bibliography again but we are not sure about what you refer. We have contacted the editor and we are waiting for her answer in order to improve this part according to the journal requirements.
Reviewer 2 Report
I think that the authors highlight a common clinical problem, regarding the reliability og AMA testing. It would be very interesting to see if liver biopsies are available whether there is a correlation between the severity of PBC and AMA positivity in various tests, but other than that the manuscript is well written and should be published.
Author Response
It would be very interesting to see if liver biopsies are available whether there is a correlation between the severity of PBC and AMA positivity in various tests, but other than that the manuscript is well written and should be published.
Thank you for your suggestion and for your time reviewing our paper. First of all, according to the current protocols, not all patients underwent biopsies. However, in some cases with biopsy, the available samples for the study were not collected at the same time of biopsy, that could have been years ago. Therefore, despite the interest of your proposal, we could not study this correlation.
Reviewer 3 Report
Arnaldos-Pérez et al. reported the discordant results of the tests for PBC.
1. Major problem is that the number of patients is too small.
2. Authors should show the biopsy-proven PBC.
3. Authors should show the details of group A.
Author Response
First of all, thank you for your time in reviewing our paper. We have carefully considered your comments and tried our best to improve our current version according to them.
- Major problem is that the number of patients is too small.
We agree with you, the number of patients is low. This is because patients are usually clearly negative or positive for AMA-M2. Only a small proportion of patients have low positive or discordant results. Our aim was to clarify whether low positive and discordant results were true positives. Therefore, we collected all the samples with low positive or discordant results (group A) over a period of 2 years and only some of the clearly positive patients (group B), which were not our main goal.
Further studies with larger sample sizes should be performed, however, our study may help clinicians and immunologists to better interpret AMA-M2 results.
- Authors should show the biopsy-proven PBC.
We have prepared supplementary tables with this information. In 3 cases biopsy was not performed due to patients met the criteria for PBC and biopsy was not necessary.
- Authors should show the details of group A.
According to your suggestion we have prepared two supplementary tables, one with immunological data and the other with clinical data. In fact, while preparing these materials, we noticed that we had made a mistake. Two patients from group A (A/14 and A/24) did not strictly met criteria for PBC and we have made the corresponding changes throughout the manuscript.
Round 2
Reviewer 3 Report
The number of patients is too small.
Author Response
The number of patients is too small.
We agree with you, the number of patients is low. However, to obtain a cohort with the characteristics of group A, we collected samples over a period of 2 years. In the future, we will pay attention to new similar patients to perform these analyses.
We also agree that further studies with larger sample sizes should be performed.